# Exploring synchronous, asynchronous, and conventional online courses in higher education

**Eunkyung Moon**[1]☻, **Won Sug Shin**[iD][2]☻*

1 Center for Teaching and Learning, Eulji University, Seongnam-si, Gyeonggi-do, South Korea,
2 Department of Korean Language Education, Incheon National University, Incheon, South Korea

☻ These authors contributed equally to this work.
* wsshin@inu.ac.kr

## Abstract

Online learning has expanded substantially in recent years, yet comparatively little is known about how different online modalities influence learners' perceptions and their intention to continue learning. This study compared three predominant online learning modalities—synchronous (Syn), asynchronous (Asyn), and conventional online courses (COC)—using an extended Information Systems Success Model (ISSM) integrating system quality (SQ), information quality (IQ), teaching presence (TP), self-efficacy (SE), perceived usefulness (PU), learning satisfaction (LS), and continuance intention (CI). The study addressed two questions: (1) how the latent means of ISSM constructs differ across Syn, Asyn, and COC, and (2) whether the structural paths among these variables differ across modalities. Survey data from 795 undergraduate students in South Korea were analyzed using latent mean analysis and multi-group structural equation modelling. Results showed that COC consistently demonstrated higher latent mean values than Syn and Asyn, underscoring the benefit of systematically designed courses. Multi-group analysis further revealed significant modality-specific variations, with relationships such as IQ→LS and PU→CI being strongest in the COC model. Overall, the findings indicate that Syn, Asyn, and COC engage ISSM mechanisms in distinct ways, highlighting the importance of modality-sensitive instructional and system design. These insights provide evidence-based guidance for enhancing the quality, satisfaction, and sustainability of online learning environments.

## Introduction

Online learning has drawn significant attention as a system designed to complement traditional education and facilitate learning beyond the limitations of time and space. From the emergence of web-based learning in the early 2000s and the expansion of MOOC-based systems in the 2010s to the global transition to online instruction during the COVID-19 pandemic, online learning has evolved beyond its initial role

**Data availability statement:** All data underlying this study have been uploaded as Supplementary information and are fully available without restriction.

**Funding:** This work was supported by the Incheon National University Research Grant in 2021 (No. 2021-0149).

**Competing interests:** The authors have declared that no competing interests exist.

as a substitute or supplement to traditional education and is now widely recognized as a fundamental instructional modality in higher education [1]. While the global transition to online instruction during the COVID-19 pandemic accelerated its adoption, the current focus has shifted towards the sustainability and quality assurance of these educational systems in the post-pandemic era. Educational institutions are now integrating various online modalities into conventional curricula to leverage their pedagogical affordances and develop systems that foster diverse and pedagogically robust learning environments [2]. However, a critical challenge remains as many instructors continue to implement these modalities without a comprehensive understanding of how to appropriately design, prepare for, and manage distinct forms of online learning [1,3].

Currently, online learning can be broadly categorized into three types: synchronous (Syn), asynchronous (Asyn), and conventional online courses (COC) [4]. Syn replicates traditional classroom settings using video conferencing tools such as Zoom and WebEx, enabling real-time interaction. In contrast, Asyn relies on learning management systems (LMS) to deliver self-recorded instructional videos, often without sufficient instructional design considerations. COC, however, are distinct in that they are designed from the outset with well-planned multimedia instruction, applying instructional design principles to produce high-quality educational content [1,5].

An expanding body of research has investigated how diverse forms of online education should be conceptualized and operationalized to achieve quality online learning, including systematic analyses of the pedagogical affordances and constraints of each modality, the instructional activities required to optimize learning processes, and the design strategies through which limitations may be mitigated and strengths amplified so that these modalities can mature into fully realized educational systems [5,6]. For example, researchers have explored various dimensions, including changes in learning activities [7], distinctions between Syn and Asyn [2,6,8], comparative analyses of face-to-face and Syn [9], shifts in instructional activities [5], and learners' satisfaction levels [3]. These studies provide valuable insights into how online learning can be refined to better align with educational goals and learners' needs.

More recent studies and practical experiences in online learning have increasingly emphasized the need to identify strategies for delivering online learning more effectively and sustainably. Central questions include whether learners consistently prefer online learning, how satisfied they are with their learning experiences, and how educational systems can support the long-term sustainability of online learning environments [5]. Despite this growing interest, existing research has yet to comprehensively examine learners' satisfaction and continuance intention across the three most representative online learning modalities—Syn, Asyn, and COC [4].

To address this gap, as well as the theoretical and methodological limitations identified in prior research, this study employs an extended Information Systems Success Model (ISSM) to systematically compare learners' perceptions across different forms of online learning. Unlike prior research that has often examined technological system quality (SQ) and pedagogical interaction in isolation, our research model integrates teaching presence (TP) and self-efficacy (SE) alongside established

ISSM factors. This integration allows for a holistic examination of how distinct instructional characteristics—such as the immediacy of interaction in Syn versus the autonomy in Asyn—interact with SQ to influence continuance intention to use (CI). Methodologically, we utilize latent mean analysis to rigorously assess differences in the perceived levels of these constructs, followed by multi-group structural equation modelling to examine whether the underlying mechanisms driving satisfaction and CI differ across modalities. Clarifying these distinct structural relationships is essential for advancing online learning from a supplementary or transitional option into a sustainable, pedagogically robust educational system. Accordingly, this study addresses the following research questions:

RQ1. To what extent do the latent means of ISSM constructs vary across Syn, Asyn, and COC?

RQ2. To what extent do the structural paths among ISSM variables differ across these modalities?

## Literature review

### Synchronous and asynchronous online learning

Synchronous online learning refers to situations in which students and instructors meet in real time despite being geographically separated [9]. This mode of learning enables immediate interaction and feedback similar to face-to-face instruction [10], facilitated through tools such as video and audio conferencing, chat, web cameras, and other real-time communication technologies [11], and students can also participate in asynchronous engagement through tools such as discussion boards. Syn enables students who experience feelings of social disconnection in online environments to engage more actively in learning [9] and receive real-time feedback, which has been shown to positively influence learning satisfaction [8] and persistence [12]. However, other researchers [9] have noted that the instructional benefits commonly attributed to Syn depend substantially on the degree of interaction incorporated into the session, with non-interactive, lecture-centered formats demonstrating comparatively limited effects on learner engagement and affective outcomes. Although Syn had not been widely adopted as a formal component of conventional online learning, it proliferated rapidly during the pandemic under the label of Emergency Remote Teaching (ERT). Yet this rapid expansion was also accompanied by criticism, particularly regarding insufficient preparation and the absence of the structured instructional design that characterizes high-quality online learning [1–3].

In contrast, Asyn is defined by its reliance on asynchronous interaction, where communication and engagement occur at different times [9,13,14]. To support this mode of learning, instructors are encouraged to provide diverse online learning materials, such as pre-recorded videos, that learners can access anytime and anywhere [15,16]. A key feature of Asyn is delayed communication, which allows participants to interact and respond at various intervals [11]. This approach enables learners to complete tasks remotely with internet access within flexible timeframes, such as a weekly schedule. Zhang et al. [8] summarized the characteristics of Asyn, stating that it encompasses activities such as watching pre-recorded lessons, viewing slide presentations, or engaging in discussions at varied times.

Building on this distinction, prior research has examined various dimensions of online learning and implemented multiple interventions aimed at improving learning performance. In general, studies have reported that Syn offers advantages related to presence, interaction, and learner satisfaction [8,12], whereas Asyn has been associated with increased motivation in collaborative writing [17], opportunities for repeated engagement with lesson materials at one's own pace, and flexible participation anytime and anywhere [15]. However, the research outcomes pertaining to the comparative effectiveness of Syn and Asyn reveal inherent complexity. For example, Nieuwoudt [4] highlighted the critical role of fostering meaningful interactions in online education, emphasizing the importance of providing abundant opportunities and options for interaction, as the pivotal contrast between Syn and Asyn predominantly lies in their approaches to interaction. However, Iglesias-Pradas et al. [18] indicated that the choice of Syn and Asyn and that of virtual communication tools did not significantly affect students' academic performance.

Taken together, the relative strengths and limitations of Syn and Asyn appear to vary depending on factors such as the learning context, instructional conditions, and the ways in which instructors design and implement each modality. Consequently, neither mode can be unequivocally prioritized over the other. These findings underscore the ongoing need for approaches and research aimed at mitigating the weaknesses and amplifying the strengths of both modalities. In this regard, it remains worthwhile to further investigate the effectiveness of these two forms of online learning. Against this backdrop, examining COC which are purposefully designed through systematic instructional processes becomes critical for understanding how well-prepared online learning environments may address the limitations observed in both Syn and Asyn.

## Conventional online courses (COC)

COC constitute a qualitatively different online learning modality, as they are deliberately designed through comprehensive instructional design processes that extend beyond the rapid or minimally structured implementations often associated with Syn and Asyn [1]. In practice, COC are implemented in at least two distinct forms: institutionally developed online courses that undergo systematic instructional design and development, and pre-developed OER-based courses that are incorporated into institutional curricula with varying degrees of pedagogical adaptation. Prior research has characterized institutionally developed COC as being intentionally designed through the integration of technological and pedagogical innovations to meet high-quality instructional standards [3]. This institutional form underscores the necessity for instructors to undergo a comparatively extensive and intricate instructional design process, ensuring the delivery of a quality learning experience that surpasses instructors' self-created content. Hodges et al. [1] explained that a full-course development project can take months as it includes full-course design support, professional development opportunities, content development, LMS training and support, and multimedia creation in partnership with faculty experts.

On the other hand, due to the difficulty of producing a sufficient number of high-quality COC, higher education institutions have attempted to effectively conduct educational activities by utilizing pre-developed online courses or open educational resource (OER) repositories, such as OpenCourseWare (OCW) and Massive Open Online Courses (MOOCs). The OCW model has served as a catalyst for expanding access to high-quality university learning resources through open dissemination, thereby institutionalizing a culture of openly shared knowledge that aligns with the broader aims of OER to advance a more equitable and globally shared educational commons [19]. In fact, significant efforts and research have been dedicated to meaningfully integrating open educational resources such as OCW into educational settings. For example, studies have explored the importance of perceived administrative support in promoting the use of OCW [20], as well as how OCW can be introduced and effectively utilized within higher education institutions [19]. However, despite the widespread implementation of online learning in higher education settings, studies investigating the incorporation of pre-developed online courses such as OCW as learning materials remain scarce, emphasizing the need for relevant studies.

Taken together, although Syn, Asyn, and COC share core characteristics as online learning modalities, they differ substantially in their instructional design and implementation processes within higher education institutions. Accordingly, understanding how learners perceive the relative value of each modality is essential for deriving evidence-based implications for the effective design and sustainable implementation of online learning environments.

## Extended information system success model (ISSM)

The effectiveness of a newly introduced information system in an organization must be validated. In an effort to demonstrate this, Delone and McLean [21] laid the groundwork for theoretical research by proposing a research model that comprehensively analysed prior studies pertaining to the success of information systems. They suggested that further development and validation are needed for their model [22]. Several researchers have since modified and tested the

DeLone and McLean [21] model. For example, Seddon [23] and Rai et al. [24] asserted the necessity of incorporating "Perceived Usefulness (PU)" into the model as they considered "System Use" to be a behavior. However, DeLone and McLean's [25] revised information system success model (ISSM), which selectively integrated diverse research findings and outcomes from prior studies on information system success, excluded PU.

Subsequently, researchers introduced additional variables to the Delone and McLean [25] model to investigate the influence of specific factors on user satisfaction and the intention to sustain usage. Wang [22] drew attention to the ongoing debate regarding the concepts of use and PU in ISSM, presenting an alternative model that once again incorporates the PU concept. Puspitarini and Ardhani [26] also contended that the updated DeLone and McLean ISSM cannot be entirely accounted for by the three technical aspects influencing it. This is because the "Information Technology Use" factor involves elements from a psychological decision perspective, thereby indicating the need to include psychological variables. According to Sabeh et al.'s [27] research on the inclusion of additional theoretical variables/key factors in IS success factor studies, there was a significant utilization of user-related factors (e.g., SE) and system quality-related factors (e.g., PU) within the original/updated Delone and McLean IS success model.

Efforts have been ongoing to elucidate the application of diverse technologies, grounded in ISSM, while simultaneously considering various exogenous variables. For example, studies examining university students' continued intention to participate in online learning revealed that PU played a critical role in the success of information systems, such as satisfaction and CI [28]. Other studies (e.g., [29,30]) enhanced the ISSM model by introducing a SE construct as a precursor to user satisfaction and usage [31]. Kishore and McLean [32] argued that SE influenced PU in the context of technology adoption. They also explained that prior to technology adoption and use, people may use their SE as an anchor to form beliefs about the usefulness outcomes from technology adoption. Islam et al. [33] showed that PU mediated the relationships between SE, satisfaction, and intention to use technology when teachers adopted new technology.

One of the most critical issues for understanding successful online learning is teaching presence, which reflects the instructional, facilitative, and organizational roles of the instructor [13]. Teaching presence emerged in the Community of Inquiry framework [34], which was designed to support the design, development, and assessment of effective online learning [35]. The role of teaching presence has recently received significant attention, especially in the online learning environment. Teaching presence describes students' perception of the accessibility, professionalism, and supportiveness of teachers [10]. Studies have reported that teaching presence significantly influenced the satisfaction [36] and persistence [37] of online learners, and it was the most crucial variable in determining course outcomes [38]. Furthermore, Zhang et al. [8] posited that an elevated level of teaching presence correlated with improved performance in Syn compared to Asyn, indicating that online learning formats are also subject to such influences. Therefore, to comprehensively evaluate learners' perceptions across Syn, Asyn, and COC, it is necessary to integrate the technological perspective of ISSM with the pedagogical perspective of TP. Considering these discussions, the model in this study extends the original ISSM by incorporating key psychological factors (SE, TP) alongside the traditional system factors of SQ and IQ. This extended framework integrates technological and psychological dimensions, providing a comprehensive foundation for examining learners' perceptions across different online learning modalities. Given the distinct instructional characteristics of the three online learning modalities, this enriched ISSM enables a systematic comparison of how modality-specific features influence PU, LS, and CI. Specifically, Syn emphasizes real-time interaction [11], rendering teaching presence particularly salient [10], whereas Asyn supports learner autonomy and are more closely associated with SE [8]. In contrast, COC, which are systematically designed with structured multimedia pathways, are more likely to enhance IQ and PU, thereby supporting instructional effectiveness [1,3] (Fig 1). Given that service quality refers to the quality of support services to system users [39], and that all participants in this study used the same LMS system within a single institutional context, service quality was not included as a separate construct in the present model.

Despite the substantial growth of online education and related research, studies specifically examining college-level students' CI within the context of higher education remain relatively limited. In particular, empirical evidence regarding how

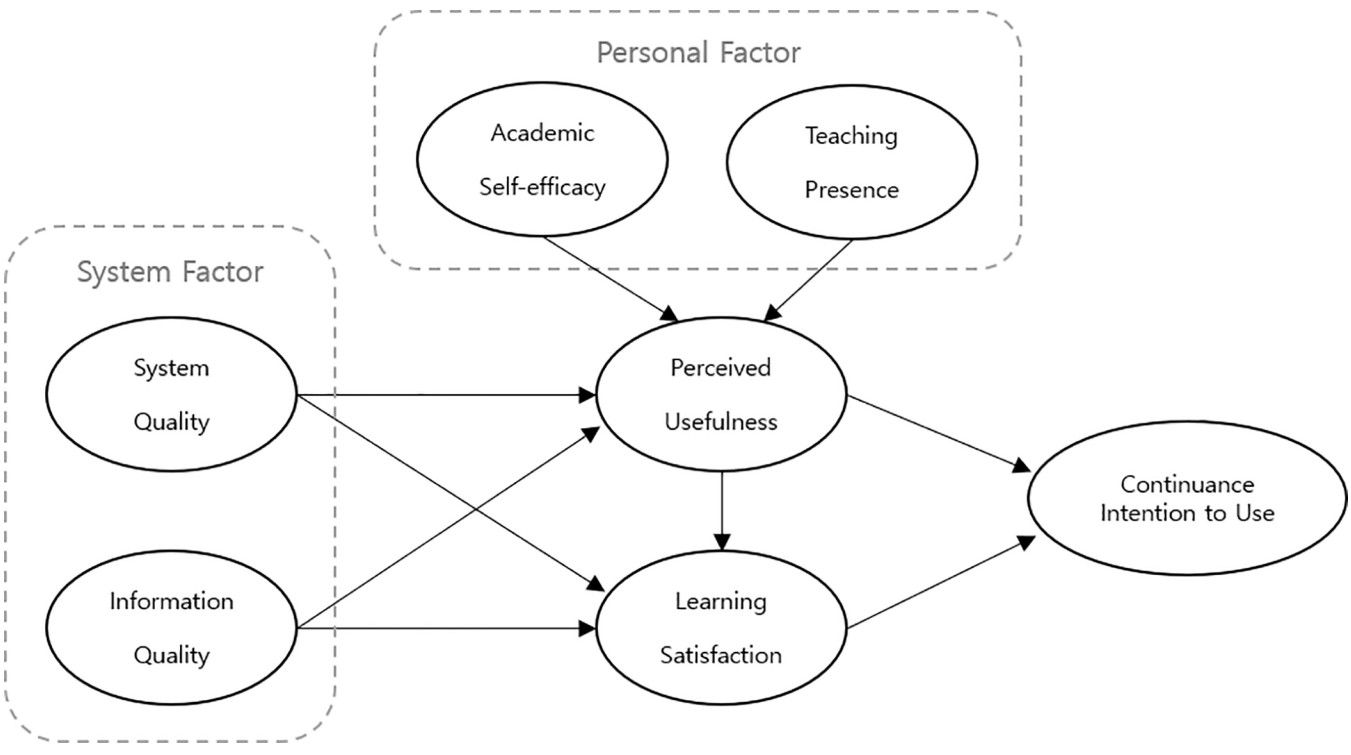

**Fig 1. Research model.**

the distinct characteristics of Syn, Asyn, and COC differentially influence learners' intention to continue usage is scarce. Consequently, this study adopts the ISSM—widely recognized as a well-established framework for examining system adoption and continuance behavior—as the primary theoretical lens to compare these learning modalities. Clarifying these mechanisms through the extended ISSM is essential for understanding how learners respond to diverse online learning environments. Therefore, the present study aims to elucidate the structural relationships underlying these three online learning types by applying an extended ISSM, thereby enhancing our understanding of learners' perceptions and CI across Syn, Asyn, and COC.

## Methodology

### Participants

This study analyzed responses from 795 undergraduate students (male, 403, 50.7%; female, 392, 49.3%) enrolled at a university in South Korea. Over a two-week period in mid-December 2022, the survey was distributed via the Learning Management System (LMS) of specific courses that were pre-identified as either Syn, Asyn, or COC. To ensure the validity of the modality classification and minimize potential confusion arising from learners' concurrent enrollment in multiple courses, participants were explicitly instructed to respond to the survey items based solely on their learning experience in the specific course where they accessed the survey link. Among the respondents, 313 students reported experience with Syn, 316 with Asyn, and 166 with COC.

This study involved a minimal-risk, non-interventional survey conducted as part of regular educational practices. In accordance with national legislation and guidelines in South Korea—including the *Bioethics and Safety Act*, which allows exemptions for anonymous, practice-based educational research that does not involve identifiable or sensitive personal

information—the study was determined to be exempt from formal Institutional Review Board (IRB) review. Therefore, IRB approval was not required. All procedures adhered to ethical standards for research involving human participants, and no personally identifiable or sensitive data were collected. All participants were informed about the purpose of the study, and participation was voluntary and based on informed consent.

## Research instruments and measures

To examine the association and the difference between the types of online learning, a survey questionnaire consisting of seven study variables and demographic questions was used. The survey questions were adapted from previous studies (Table 1). The seven validated measures used in the research model were as follows: system quality (SQ), information quality (IQ), academic self-efficacy (SE), teaching presence (TP), perceived usefulness (PU), learning satisfaction (LS), and continuance intention to use (CI).

SQ refers to the information system's usability, accessibility, utility, complexity, and response time [25]. In this study, accessibility, stability, response time, and ease of use are considered to be the important aspects of a good system. IQ was described by Delone and McLean as the level of output produced by an information system and is a crucial aspect of determining the efficacy of information [47]. Questionnaire items on IQ included those regarding system usefulness, understandability, interestingness, reliability, completeness, timeliness. SE refers to the subjective belief in one's ability to successfully learn or perform given learning tasks [48], which encompasses "self-efficacy for learning" and "self-efficacy for performance." TP was assessed using 15 items, encompassing systematic lesson execution by teachers, facilitation and assessment of learning, as well as the concept of instructor presence. PU was evaluated using items that inquired about the improvement of learners' learning outcomes, and effectiveness and efficiency regarding online learning. LS was assessed using questions on learning quantity and value, and whether respondents encouraged others to take the same online course. CI included questionnaire items on preference, continuous intention, and expanding and recommending online learning.

The 48 items in the questionnaire were measured on a 5-point Likert scale. As all respondents were Korean, the original survey items written in English were first translated into Korean by researchers, then reviewed by three bilingual PhDs in educational technology to ensure content validity. The Cronbach's alpha of the scale was 0.97 in the present study.

## Data analysis

To comprehensively examine the differences between Syn, Asyn, and COC, this study adopted a sequential analytic framework spanning measurement invariance, latent mean analysis (LMA), and multi-group structural equation modelling (SEM). This approach was chosen to ensure a rigorous comparison: first, measurement invariance tests confirmed that constructs were conceptually equivalent across groups; next, LMA identified differences in the magnitude of learners' perceptions; and finally, multi-group SEM revealed variations in the underlying structural mechanisms. This integrated

**Table 1. Measures.**

| Variables | Items | Sample question | Cronbach's α | Source |
|---|---|---|---|---|
| SQ | 4 | Online learning is easy to use. | .881 | Yakubu and Dasuki [40] |
| IQ | 6 | The information provided by Online learning is useful. | .915 | Urbach, Smolnik, and Riempp [41] |
| SE | 8 | I am confident in my ability to perform well on exams for this subject. | .916 | Bong et al. [42] |
| TP | 15 | The instructor conducted this course passionately. | .929 | Koh [43] |
| PU | 3 | I think using Online learning can increase the effectiveness of my studies. | .891 | Liu [44] |
| LS | 8 | I would encourage others to take the courses Online learning provides. | . 938 | Shin [45] |
| CI | 4 | I think Online learning should be implemented in other classes. | .933 | Lee et al. [46] |

strategy allows for simultaneously examining both the quantitative levels and qualitative pathways distinguishing the three types. Prior to these main analyses, normality tests were conducted to ensure that the data met the necessary assumptions. To make cross-group comparisons, the latent mean analysis approach was used for latent variables that could not be directly measured. This approach has been recommended for cross-group comparisons while the variables are conceptualized as latent constructs that could not be directly measured [49].

Measurement invariance is a prerequisite for making cross-group comparisons using latent mean analysis [49]. Before testing the group differences in the means of latent variables, the configural, metric, and scalar invariance must be examined to evaluate the equivalence of the groups [49]. Finally, measurement invariance tests were conducted to determine latent mean differences across the three groups [50–52].

Next, a structural model invariance test was employed across the groups to assess the significance of differences in each path. This involved comparing a series of models with different constrained paths to determine if the responses across the groups were equivalent. Testing the structural model invariance across the groups involved two main procedures: metric invariance equality constraints [53], which examined the differences in paths between the groups. As a precondition to compare the structural path coefficient differences, metric invariance equality tests were conducted in subsequent steps using the same baseline research model. This process involved conducting a test of the unconstrained model followed by the equal factor loading model and the equal path model.

SEM was used to examine the hypothetical model and was employed using maximum likelihood estimation to test the measurement model's fitness. Analyses were performed using AMOS version 21. $\chi^2$ test was employed to evaluate the model fit. To evaluate model fit, we used the Comparative Fit Index (CFI), Tucker-Lewis Index (TLI), and Root Mean Square Error of Approximation (RMSEA). Following established guidelines, CFI and TLI values greater than 0.90 were considered indicative of a good fit [54]. For RMSEA, values less than 0.05 indicate a close fit, while values between 0.05 and 0.08 are considered reasonable errors of approximation [55].

## Results

Before presenting the detailed results, we briefly clarify how each research question was addressed in the analyses. RQ1 was examined through the latent mean analysis, which compared perceived levels of the extended ISSM constructs across Syn, Asyn, and COC. RQ2 was addressed through the multi-group structural equation modeling, which tested whether the structural relationships among these constructs differed by learning modality.

### Correlation, means and standard deviations

Table 2 presents the correlations and descriptive statistics of the seven measured variables, which were significantly positive correlated (r = .343 to.822, p < .001). Both convergent validity and discriminant validity were satisfied [56]. Univariate normality was confirmed based on skewness values ranging from −0.617 to −0.108 and kurtosis values ranging from −0.571 to 0.447, all falling within the recommended thresholds of |3| and |10|, respectively [56].

### Latent mean analysis

Several invariance tests were performed before conducting the latent mean analysis. The baseline configural, metric, and scalar invariance models all fit well to the data (Table 3). First, the configural invariance was assessed without constraining equality across the groups (M1). The goodness-of-fit results ($\chi^2$ = 1107.49, CFI = 0.959, TLI = 0.948, RMSEA = 0.039) indicated similar structural patterns across the groups. This implies that the configural model can serve as a baseline for comparison with other restricted models in the invariance tests. Subsequently, metric invariance was assessed by constraining the factor loadings to be equal across the groups (M2). The results demonstrated a good model fit ($\chi^2$ = 1153.276, CFI = 0.957, TLI = 0.949, RMSEA = 0.038). Finally, a scalar invariance test was performed by constraining the intercepts across the groups to be invariant (M3). The model fit indices showed that $\chi^2$ = 1553.906, CFI = 0.938,

Table 2. Correlation, means and standard deviations (N = 795).

| | 1 | 2 | 3 | 4 | 5 | 6 | 7 |
|---|---|---|---|---|---|---|---|
| 1 SQ | 1 | 0.578*** | 0.364*** | 0.429*** | 0.381*** | 0.394*** | 0.343*** |
| 2 IQ | | 1 | 0.66*** | 0.777*** | 0.631*** | 0.698*** | 0.508*** |
| 3 SE | | | 1 | 0.715*** | 0.633*** | 0.676*** | 0.511*** |
| 4 TP | | | | 1 | 0.643*** | 0.702*** | 0.505*** |
| 5 PU | | | | | 1 | 0.822*** | 0.768*** |
| 6 LS | | | | | | 1 | 0.799*** |
| 7 CI | | | | | | | 1 |
| Skewness | −0.617 | −0.468 | −0.252 | −0.108 | −0.412 | −0.366 | −0.439 |
| Kurtosis | 0.089 | 0.447 | −0.158 | −0.066 | −0.402 | −0.386 | −0.571 |
| M | 3.79 | 3.85 | 3.53 | 3.62 | 3.53 | 3.54 | 3.51 |
| SD | 0.904 | 0.747 | 0.757 | 0.697 | 1.003 | 0.920 | 1.101 |

*Note.* *p < .05; ** p < .01; *** p < .001.

Table 3. Fit indices for invariance verification.

| Model | | χ2 (df) | CFI | TLI | RMSEA | Δχ2 (Δdf) | ΔCFI | ΔTLI | ΔRMSEA |
|---|---|---|---|---|---|---|---|---|---|
| M1 | Configural | 1107.49 (504) | 0.959 | 0.948 | 0.039 | | | | |
| M2 | Metric | 1153.276 (532) | 0.957 | 0.949 | 0.038 | 45.786 (28) | .002 | .001 | .001 |
| M3 | Scalar | 1553.906 (574) | 0.938 | 0.926 | 0.046 | 400.630 (42) | .019 | .023 | .008 |

TLI = 0.926, and RMSEA = 0.046. Given that the assumptions of configural, metric, and scalar invariance were satisfied; ΔCFI<0.02 [50], ΔTLI<0.05 [51], ΔRMSEA< 0.015 [52], a latent mean analysis could be conducted.

Latent mean analysis was employed to estimate the mean differences across the three groups. COC were selected as the primary reference group (fixed to zero [49]), as COC represents the standard of online instruction designed with systematic instructional principles, serving as a baseline for comparing the distinct characteristics of Syn and Asyn. Furthermore, to comprehensively examine the differences between all three modality pairs (including Syn vs. Asyn), we systematically rotated the reference group. This procedure allowed us to obtain precise estimates and statistical significance for the mean differences among all three groups, as presented in Table 4. Overall, COC had higher mean values for all factors than Syn and higher mean values for six variables than Asyn except for system quality. Asyn had higher mean values for SQ, SE, and CI than Syn, whereas Syn had higher mean values for TP than Asyn.

## Structural model invariance test across the groups

The structural model invariance test across the groups assesses whether the proposed theoretical model is parameter invariant across groups. The SEM of the hypothesized model on the whole sample was performed before conducting the multi-group analysis. The results of the SEM showed that the proposed model for the whole model yielded a good fit, χ2 = 776.792, CFI = .961, TLI = .953, RMSEA = .066 (Table 5). Subsequently, the multi-group analysis structural equation model analysis was conducted.

The structural model invariance across the groups was analyzed by testing the metric invariance equality constraints, which examined whether the responses across the groups were identical in the research model, and the cross-group

## PLOS One

**Table 4. Results of latent means analyses.**

| | COC#/ Syn | | COC#/ Asyn | | Syn#/ Asyn | |
| --- | --- | --- | --- | --- | --- | --- |
| | Estimate | Cohen's d | Estimate | Cohen's d | Estimate | Cohen's d |
| SQ | −0.311** | −0.377 | 0.008 | 0.010 | 0.319*** | 0.387 |
| IQ | −1.102*** | −0.628 | −1.052*** | −0.599 | 0.05 | 0.028 |
| SE | −2.038*** | −0.424 | −1.679*** | −0.350 | 0.359* | 0.075 |
| TP | −3.286*** | −0.268 | −3.968*** | −0.323 | −0.682* | −0.056 |
| PU | −2.340*** | −2.574 | −2.029*** | −2.232 | 0.311 | 0.342 |
| LS | −0.783*** | −0.143 | −0.671*** | −0.122 | 0.112 | 0.020 |
| CI | −0.900*** | −0.802 | −0.671*** | −0.598 | 0.229** | 0.204 |

*Note.* *p < .05; ** p < .01; *** p < .001.

The latent mean values for (#) reference group were set to zero.

equality constraints, which examined whether there were differences in the paths across the groups. This enabled us to determine whether the samples from the three groups yielded consistent results within the same research model and to analyze any differences exhibited by the research model across the groups.

First, all parameters in the model were released among the groups, and M1 was obtained ($\chi2 = 1188.074$, CFI = 0.954, TLI = 0.945, RMSEA = 0.04). Second, the factor loads were equalized among the groups and M2 was obtained ($\chi2 = 1236.435$, CFI = 0.953, TLI = 0.946, RMSEA = 0.04). Lastly, the paths between the latent variables were equalized among the groups, and M3 was obtained ($\chi2 = 1276.593$, CFI = 0.951, TLI = 0.946, RMSEA = 0.04). Subsequently, M2 and M1, M3 and M2 were compared (Table 5). The fit statistics for all three models were acceptable, as they consistently satisfied the recommended benchmark criteria (CFI, TLI > 0.90; RMSEA < 0.08). When the fit statistics of M2 and M1 were compared, it was found that they did not change significantly ($\Delta$CFI = 0.001, $\Delta$TLI = −0.001, $\Delta$RMSEA = 0). Finally, analysis of the change in the fit statistics of M3 and M2 revealed that the change of the fit values were not significant ($\Delta$CFI = .002, $\Delta$TLI = 0, $\Delta$RMSEA = 0). Therefore, the paths in the model were valid for the three types of online learning.

Second, cross-group equality constraints were tested to examine the significant differences in the relationships among the variables of the research model across the groups. Nine models, each imposing equality constraints on the path coefficients within the model, were compared with the baseline model that did not impose equality constraints on any path coefficients (Table 6). The results revealed four significant path differences in the model between the groups. They also indicated that the paths from IQ to PU (P2; $\Delta\chi2 = 9.334$, p = 0.009), IQ to LS (P6; $\Delta\chi2 = 7.236$, p = 0.027), PU to LS (P7; $\Delta\chi2 = 14.47$, p = 0.001), and PU to CI (P9; $\Delta\chi2 = 7.03$, p = 0.03) were significantly different between the three groups.

Table 7 shows the standardized regression weights of the paths for each type of online learning. Consistently, the paths from IQ to LS (P6), PU to LS (P7), LS to CI (P8), and PU to CI (P9) exhibited significant influence across all three groups. However, SQ did not have a significant influence on PU (P1) and LS (P5). In contrast, the paths from IQ (P2), SE (P3), and TP (P4) to PU exhibited different degrees of influence across the three groups. After testing cross-group equality constraints, the results showed that the effects of the paths in the model differed across the three groups (Fig 2).

**Table 5. Summary statistics for tested models in multi-group analyses.**

| Model | | χ2 | df | CFI | TLI | RMSEA |
| --- | --- | --- | --- | --- | --- | --- |
| The hypothesized model | | 776.792 | 174 | 0.961 | 0.953 | 0.066 (.037−.043) |
| M1 | Unconstrained | 1188.074 | 522 | 0.954 | 0.945 | 0.04 (.037−.043) |
| M2 | Factor loading equal | 1236.435 | 550 | 0.953 | 0.946 | 0.04 (.037−.043) |
| M3 | Paths equal | 1276.593 | 568 | 0.951 | 0.946 | 0.04 (.037−.043) |

**Table 6. Fit indices for different models with constraints on each path.**

| Paths | | | | △df | △χ2 | p | △TLI |
|---|---|---|---|---|---|---|---|
| P1 | SQ | → | PU | 2 | 2.323 | 0.313 | 0 |
| P2 | **IQ** | → | **PU** | **2** | **9.334**\*\* | **0.009** | **0** |
| P3 | SE | → | PU | 2 | 3.053 | 0.217 | 0 |
| P4 | **TP** | → | **PU** | **2** | **2.449** | **0.294** | **0** |
| P5 | SQ | → | LS | 2 | 2.065 | 0.356 | 0 |
| P6 | **IQ** | → | **LS** | **2** | **7.236**\* | **0.027** | **0** |
| P7 | PU | → | LS | 2 | 14.47\*\* | 0.001 | 0.001 |
| P8 | **LS** | → | **CI** | **2** | **5.766** | **0.056** | **0** |
| P9 | PU | → | CI | 2 | 7.03\* | 0.03 | 0 |
| All path | | | | **18** | **40.158**\*\* | **0.002** | **0** |

*Note.* \*p < .05; \*\* p < .01; \*\*\* p < .001.

**Table 7. Path coefficients of the multi-group analyses.**

| Paths | | Syn | | | Asyn | | | COC | | |
|---|---|---|---|---|---|---|---|---|---|---|
| | | b | S.E. | β | b | S.E. | β | b | S.E. | β |
| P1 | SQ→PU | 0.104 | 0.103 | 0.088 | −0.106 | 0.095 | −0.097 | 0.013 | 0.058 | 0.017 |
| P2 | IQ→PU | −0.074 | 0.115 | −0.097 | 0.296\*\* | 0.111 | 0.42 | 0.424\*\* | 0.132 | 0.602 |
| P3 | SE→PU | 0.165\*\* | 0.056 | 0.347 | 0.153\*\*\* | 0.036 | 0.357 | 0.046 | 0.054 | 0.098 |
| P4 | TP→PU | 0.126\* | 0.055 | 0.421 | 0.028 | 0.041 | 0.099 | 0.027 | 0.043 | 0.1 |
| P5 | SQ→LS | −0.021 | 0.067 | −0.017 | −0.132 | 0.07 | −0.116 | −0.021 | 0.037 | −0.029 |
| P6 | IQ→LS | 0.123\* | 0.048 | 0.157 | 0.225\*\*\* | 0.057 | 0.309 | 0.339\*\*\* | 0.06 | 0.493 |
| P7 | PU→LS | 0.855\*\*\* | 0.054 | 0.827 | 0.73\*\*\* | 0.059 | 0.708 | 0.49\*\*\* | 0.078 | 0.502 |
| P8 | LS→CI | 0.714\*\*\* | 0.134 | 0.63 | 0.629\*\*\* | 0.095 | 0.591 | 0.279\* | 0.14 | 0.245 |
| P9 | PU→CI | 0.303\* | 0.14 | 0.258 | 0.316\*\* | 0.098 | 0.287 | 0.738\*\*\* | 0.142 | 0.662 |

*Note.* \* p < .05; \*\* p < .01; \*\*\* p < .001.

# Discussion

This discussion is structured around the two research questions guiding the study. Specifically, RQ1 is addressed by interpreting the latent mean differences across learning modalities, whereas RQ2 is discussed through the modality-specific structural relationships identified in the multi-group analysis.

## Latent mean differences across modalities

**COC versus (A)Syn.** The latent mean analysis revealed a consistent pattern in which COC demonstrated higher perceived levels of IQ, TP, SE, PU, LS, and CI than both Syn and Asyn. This result can be interpreted to mean that learners tend to evaluate systematically designed, purposefully developed online courses more favorably than formats that rely primarily on instructor-generated content or minimally structured recordings. This finding is in line with that of previous research showing that the effectiveness of online education is strengthened through carefully designed and well-organized instruction [1,16]. From the perspective of the ISSM, this pattern suggests that systematically organized course design enhances learners' quality perceptions, which subsequently contribute to PU and, in turn, to CI.

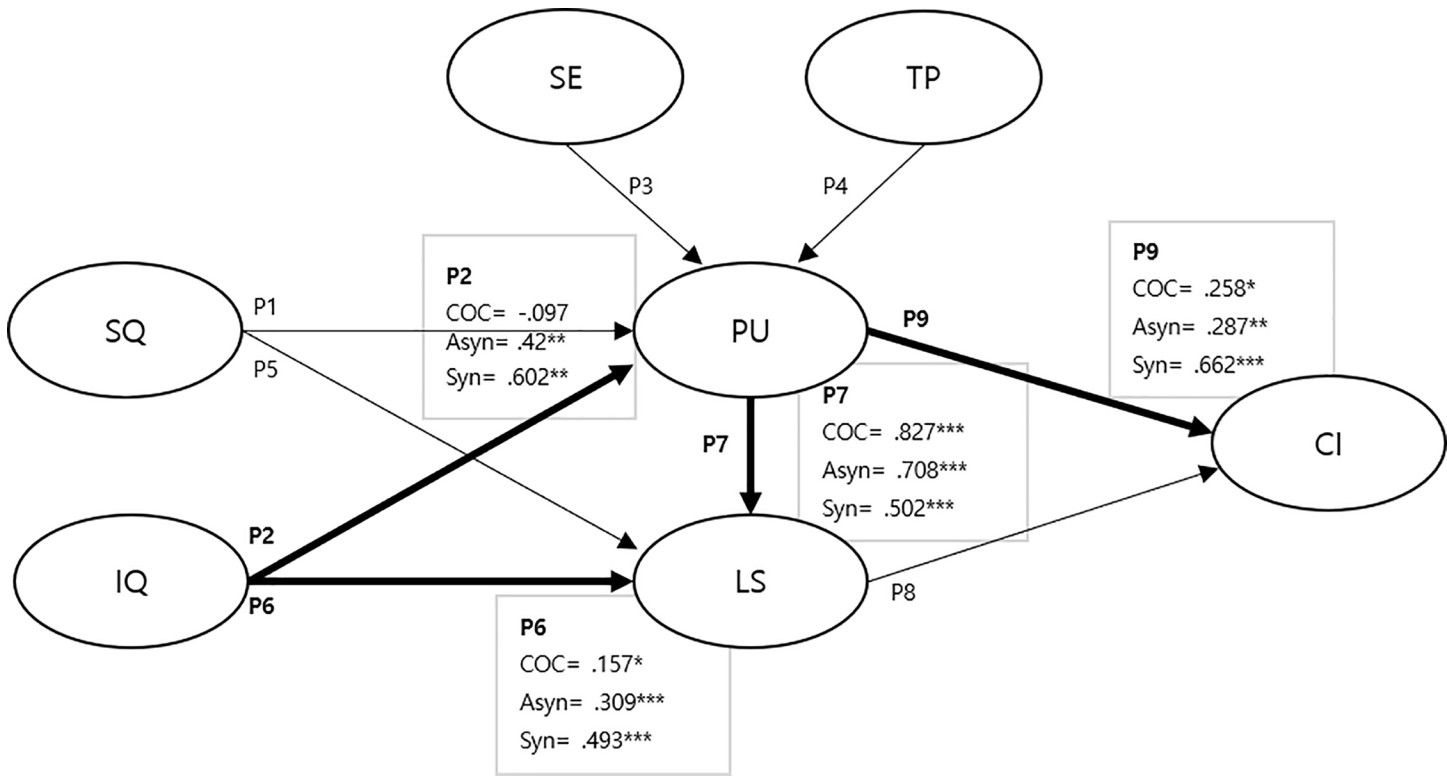

**Fig 2. The standardized coefficients for path with the significant differences between groups.** *Note.* * p<.05; ** p<.01; *** p<.001.

Although COC showed significantly higher SQ than Syn, no significant difference was observed between COC and Asyn. This implies that, considering the inherent characteristics of real-time video conferencing systems, Syn may encounter interruptions or abrupt transitions [7], whereas Asyn, which involves streaming recorded videos, may be perceived as having no notable distinctions to COC from learners' standpoint.

**Syn versus Asyn.** This study confirmed that SQ, SE, and CI were significantly higher in Asyn than in Syn. First, the higher SQ observed in Asyn can be attributed to its reliance on recorded content delivery, which enables more stable video streaming and reduces class disruptions commonly associated with real-time environments. Consistent with this interpretation, prior research has shown that learners have reported Syn to be constrained by network instability, limited interaction, and reduced concentration [7].

Second, SE was higher in Asyn. This aligns with the result of prior research indicating that students with higher SE exhibited more organized learning and self-management abilities [57], enabling flexible learning at their own pace [6]. Students with higher SE exerted more effort for academic performance and demonstrated greater engagement in learning activities [57]; therefore, students may engage in more proactive learning activities, such as repeated practice, in Asyn [8].

Third, CI was also higher in Asyn. This finding is consistent with prior research indicating that the flexibility of Asyn enables repeated and self-paced engagement with learning content [6,8,15,16], which in turn fosters learners' intention to continue participating in online courses.

In contrast, TP was significantly higher in Syn than in Asyn, indicating that interactions between instructors and learners were relatively more active in Syn than in Asyn. This finding resonates with that in existing studies indicating that Syn is more favorable than Asyn in cultivating learners' presence [9], as instructors in Syn can more clearly explain course content and organize real-time in-class discussions [8]. As such, the importance of TP in online

education has been well-established through numerous studies but further research to ensure TP even in Asyn is necessary. For example, Um and Jang [36] identified TP as a crucial factor in Syn, suggesting the need for further research on enhancing TP in online settings, while Martin et al. [9] emphasized the effective utilization of Syn and Asyn characteristics to enhance TP in online courses, advocating for diverse facilitating practices such as moderating discussions and providing feedback.

Taken together, these latent mean differences suggest that learners perceive Syn, Asyn, and COC in systematically different ways across key ISSM constructs. However, mean-level comparisons alone cannot clarify how these perceptions are dynamically related, necessitating further examination of the underlying structural relationships across modalities.

## Structural path invariance (Multi-group analysis) across modalities

To move beyond mean-level differences and examine whether the underlying ISSM mechanisms operated similarly across modalities, a multi-group structural equation model was conducted. A few statistically significant differences were observed between the types of online learning according to the model tested with multi-group analysis. The path coefficients between IQ and LS, PU and CI, PU and LS, and IQ and PU showed significant differences. The relationship between IQ and LS, PU and CI, and PU and LS were thus significant in all three models, whereas that between IQ and PU was not significant in the Syn model.

Firstly, the coefficients between IQ and LS showed that COC was the highest, followed by Asyn and Syn (COC = 0.493, Asyn = 0.309, Syn = 0.157). This implies that the level or presentation of information provided in Syn may not have adequately met learners' demands. Previous research has emphasized the importance of IQ—such as providing students with sufficient, concise, and clear information, as well as updated and engaging learning content—in enhancing overall LS and CI in online learning environments [58]. Additionally, the delivery method or structure of information in Syn may not have sufficiently contributed to learners' overall satisfaction [59]. From an ISSM perspective, this finding suggests that in learning environments where instructional explanations are primarily delivered through pre-structured content—as is typical in COC—IQ plays a central role in shaping learner satisfaction. In contrast, in Syn, limitations in IQ may be less salient due to the presence of immediate instructor interaction and real-time clarification.

Second, the relationship between PU and CI also showed that COC coefficients were significantly higher than Syn and Asyn. This suggests that PU plays a more influential role in shaping CI within the COC. Consequently, to support online learning continuity, instructional design approaches characteristic of COC may enhance learners' perceptions of online learning usefulness [1,3]. This underscores the importance of clear and well-structured instructional information in strengthening learners' PU and supporting CI in COC environments.

However, the relationship between PU and LS exhibited coefficients opposite to those observed in the PU to CI relationship. Specifically, Syn ranked the highest, followed by Asyn and COC. These contrasting patterns imply that the function of PU differs by modality. In COC, the structured and well-designed nature of the course environment strengthens the ISSM mechanism in which PU directly promotes CI. In contrast, the Syn setting, characterized by real-time interaction and instructional immediacy, reinforces the experiential component of learning, thereby amplifying the influence of usefulness on LS rather than on CI. Such findings highlight the necessity for ongoing research in online education on how to enhance meaningful interaction and ensure learning satisfaction for students.

In the Syn model, the relationship between IQ and PU was not significant, whereas the paths in the models of Asyn and COC were. This implies the need to enhance IQ in future Syn learning environments. Higher IQ is crucial for learners to perceive the usefulness of online learning, thereby increasing LS and their CI [47]. Moreover, in terms of IQ, the quality of information provided in well-designed COC is likely to be higher than that in Asyn, where instructional design processes may be less systematically applied.

According to the ISSM framework, learners' CI is shaped by a sequential mechanism in which quality perceptions (e.g., IQ, SQ) and psychological variables (e.g., TP as an extended factor) influence PU, which in turn affects LS and ultimately

CI. The patterns observed in this study should therefore be interpreted not merely as descriptive differences but as modality-dependent variations in the operation of the underlying ISSM mechanism across online learning formats.

In addition to the significant structural paths, several non-significant or relatively weaker relationships merit brief consideration. These patterns may reflect modality-specific instructional characteristics rather than theoretical inadequacy. For example, in Syn environments, real-time interaction and instructor immediacy may reduce learners' reliance on structured informational cues, thereby attenuating the influence of IQ on PU. Conversely, TP significantly influenced PU only in the Syn model, highlighting the unique value of live instructor facilitation. In addition, the non-significant influence of SQ across all groups, potentially coupled with ceiling effects in highly structured COC, may have constrained variability in certain system-related perceptions. Such findings suggest that the operation of ISSM mechanisms is contingent on instructional modality and contextual design features, underscoring the importance of modality-sensitive interpretation rather than uniform expectations across online learning formats.

## Limitations and suggestions for further work

Although this study provides meaningful empirical insight into the distinct mechanisms through which Syn, Asyn, and COC influence learners' perceptions and CI, there are several limitations to be considered when interpreting the research results.

Firstly, this study relied on self-reported survey data. Although the constructs used in the extended ISSM framework are widely validated, incorporating objective behavioral data such as log data and interaction patterns, as well as qualitative evidence through interviews or case studies, would provide a more holistic understanding of how learners experience and evaluate each modality.

Second, while the extended ISSM framework incorporated both system-related and psychological variables, additional influential factors—such as learning styles and motivation—may also shape learners' perceptions and behavioral intentions. Future research should consider expanding the model to incorporate such factors to capture more nuanced mechanisms.

Third, the survey was administered based on learners' retrospective evaluations of previously completed courses. Consequently, their responses may have been affected by memory bias or accumulated experiences with past online learning environments. Longitudinal research designs that track learners over time within each modality would provide more accurate evidence regarding modality preferences, satisfaction patterns, and CI.

Fourth, the effectiveness of each modality may vary depending on course design quality, learning activities, education level, and learner characteristics. Future studies should adopt more fine-grained analytical approaches to examine how instructional design, pedagogical strategies, and learner profiles interact with modality.

Finally, further research is warranted to investigate how modality-sensitive design strategies can enhance learner engagement, meaningful interaction, and satisfaction in online learning environments.

## Conclusions and implications

To the best of our knowledge, this is one of the few studies to systematically compare Syn, Asyn, and COC using an extended ISSM framework. Research examining learners' perceptions of different online learning modalities remains scarce, as many studies have generalized online learning under the umbrella term "e-learning" without differentiating among distinct formats. Thus, the findings of this study provide foundational background on the progression of online learning across different formats and its impact on how various online learning modalities have developed and how they uniquely shape learners' perceptions and CI. These insights can inform the development of future policies and practical guidelines for designing effective and sustainable online learning environments.

The findings of this study suggest that educational stakeholders should prioritize the development of well-prepared online learning courses that are meticulously designed rather than relying on instructor-generated content without proper

 

preparation. Enhancing instructors' course design competencies [9] is essential for ensuring high-quality learning experiences and fostering stable TP [60]. Moreover, consistent with previous research indicating that modality selection should be aligned with course characteristics and learner profiles—and that educators should critically evaluate the strengths and limitations of Syn and Asyn while investing greater effort in instructional design to optimize learning outcomes [2], the present study reinforces this perspective by demonstrating modality-specific mechanisms that require differentiated instructional and design approaches. In particular, the strong PU to CI relationship observed in the COC indicates the value of improving the clarity and structural organization of instructional information and system usability to further strengthen learners' PU and promote their CI.

In addition, existing research has shown that Syn often leads to higher levels of learner engagement and persistence through immediate feedback, real-time interaction, and heightened social presence, while autonomous academic motivation positively influences both engagement and CI across modalities through the mediating role of engagement. When considered alongside the current findings, these insights underscore the importance of strategically integrating synchronous elements—even within predominantly Asyn or structured COC environments—to enhance learners' affective involvement, support sustained participation, and strengthen the overall learning experience.

Finally, the creation of effective online learning environments requires modality-sensitive instructional strategies. In practice, for example, this means providing structured design templates for COC courses and incorporating synchronous interactions into asynchronous lessons to enhance instructional coherence across modalities. Ensuring both synchronous and asynchronous flexibility [6] while advancing technology-enabled learning infrastructure [59] will be key to providing consistent, high-quality digital academic experiences. Collectively, these findings highlight the importance of modality-sensitive design strategies that leverage the strengths of each format to support LS, engagement, and sustained participation.

## Supporting information

**S1 Dataset. Online learning dataset.** Dataset containing the survey responses (n = 795).
(XLSX)

**S1 File. Data description.** Detailed description of the variables and data coding.
(DOCX)

## Author contributions

**Conceptualization:** Eunkyung Moon.

**Data curation:** Eunkyung Moon.

**Formal analysis:** Eunkyung Moon.

**Funding acquisition:** Won Sug Shin.

**Investigation:** Eunkyung Moon.

**Methodology:** Eunkyung Moon.

**Project administration:** Won Sug Shin.

**Writing – original draft:** Eunkyung Moon, Won Sug Shin.

**Writing – review & editing:** Won Sug Shin.

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
