## [Decision Letter · Decision Letter 0]

16 Nov 2025

Dear Dr. shin,

Thank you for submitting your manuscript to PLOS ONE. After careful consideration, we feel that it has merit but does not fully meet PLOS ONE’s publication criteria as it currently stands. Therefore, we invite you to submit a revised version of the manuscript that addresses the points raised during the review process.

We look forward to receiving your revised manuscript.

Kind regards,

Da Yan, Ph.D.

Guest Editor

PLOS ONE

Journal Requirements:

3. In the online submission form, you indicated that all data are available from the corresponding author upon request.

The Incheon National University Research Grant in 2021(#2021-0149)

5. Please amend your authorship list in your manuscript file to include author Won shin.

6. Please amend the manuscript submission data (via Edit Submission) to include author Won Sug Shin

7. Please upload a copy of Supporting Information Figure S1, which you refer to in your text on page 27.

Reviewers' comments:

Reviewer's Responses to Questions

**Comments to the Author**

1. Is the manuscript technically sound, and do the data support the conclusions?

Reviewer #1: Yes

Reviewer #2: Partly

2. Has the statistical analysis been performed appropriately and rigorously?

Reviewer #1: I Don't Know

Reviewer #2: Yes

3. Have the authors made all data underlying the findings in their manuscript fully available?

Reviewer #1: No

Reviewer #2: No

4. Is the manuscript presented in an intelligible fashion and written in standard English?

Reviewer #1: Yes

Reviewer #2: Yes

Reviewer #1: Dear author,

The paper is interesting. The following are necessary for the publication of the paper:

1.The title should be re-phrased to something like “Exploring the Synchronous, asynchronous, and conventional online courses in higher education”.

2.The study research design, research questions/objectives and methodology are not stated in the abstract.

3.Five or more keywords should be added as there is no keywords list after the abstract.

4.Copy editing is recommended.

5.In the Methodology section, within the Participants subsection, lines 226-7, the author stated that the whole number of respondents is 795, showing “313 students reported experience with Syn, 316 with Asyn, and 166 with 227 COC”. Aren’t there cases where some respondents experienced more than one mode? Is it really they experienced ONLY one mode of learning? The whole world experienced the three modes, didn’t they?

6.In line 279, there is a typing error.

7.Titles of table captions are not to end with a full stop which is usually used to end a full sentence.

8.It is highly suggested to have a section titled ’Limitations and suggestions for further work’ before the Conclusion section.

9.The conclusion section is weak in the sense that it does nor truly conclude the present study. The author is invited to re-work on this bringing the main findings of this study.

10.There is no reason why Figure 2 presented before figure 1 at the end of the paper.

Best of luck.

Reviewer #2: Overall Evaluation:

The manuscript addresses a timely and relevant topic on online learning modes. The overall structure is sound; however, several sections could be improved to enhance theoretical rigor, methodological transparency, and practical relevance.

1. Introduction

Since COVID-19 is no longer a pressing issue in online education, the introduction should focus on the broader development and sustainability of online learning rather than positioning the pandemic as the central rationale.

2. Literature Review

Some of the cited references are relatively outdated (e.g., Skylar, 2009; Sun & Wu, 2016), and it is recommended that the authors include more recent studies.

The discussion of synchronous, asynchronous, and conventional online courses remains largely descriptive, without synthesizing the literature to highlight a clear research gap.

The review does not explicitly state the research questions or explain how this study addresses the theoretical or methodological limitations identified in previous work.

3. Data Analysis

The logical connection between latent mean analysis (LMA) and multi-group SEM remains unclear. Although the analytical sequence is correct, the authors should specify which group served as the reference in LMA and elaborate on how the three analytical steps are theoretically linked.

While the authors report CFI, TLI, and RMSEA values and mention ΔCFI/ΔTLI/ΔRMSEA thresholds, the actual Δ values and references for these cut-offs are not provided. Including a summary table and relevant citations (Cheung & Rensvold, 2002; Chen, 2007) would enhance transparency.

Finally, please clarify whether the reported fit indices meet accepted benchmarks (CFI/TLI > 0.90, RMSEA < 0.08) and consider reporting standard errors or confidence intervals for path coefficients to strengthen interpretability.

4. Discussion and Conclusion

The discussion section interprets the findings largely from an empirical perspective but does not fully draw on the Information System Success Model (ISSM) and its extended variables to explain the underlying mechanisms. The authors are encouraged to explicitly link the observed relationships (e.g., stronger PU→CI and IQ→LS paths in the COC mode) to the ISSM framework to enhance theoretical coherence.

Some explanations (e.g., “teaching presence increases satisfaction in the Syn mode” or “COC performs best overall”) extend beyond what is directly supported by the statistical evidence.

Although the conclusion summarizes the main implications, the discussion remains rather general. The authors could make the implications more specific and actionable, for example, by suggesting how improved information integration and system usability could strengthen the PU→CI relationship observed in the COC mode.

**Do you want your identity to be public for this peer review?** For information about this choice, including consent withdrawal, please see our For information about this choice, including consent withdrawal, please see our Privacy Policy .

Reviewer #1: **Yes:** Nawal Fadhil AbbasNawal Fadhil Abbas

Reviewer #2: No

---

## [Author Response · Author response to Decision Letter 1]

15 Dec 2025

Response to Reviewer

Reviewer #1:

1. The title should be re-phrased to something like “Exploring the Synchronous, asynchronous, and conventional online courses in higher education”:

: Thank you for your comment and offering an attractive title of the manuscript. The title has been revised accordingly to:

“Exploring synchronous, asynchronous, and conventional online courses in higher education.”

2. The study research design, research questions/objectives and methodology are not stated in the abstract:

: We agree with this comment. The abstract has been substantially revised to explicitly state the research design, key research questions, analytical methods (latent mean analysis and multi-group analysis), and sample characteristics.

3. Five or more keywords should be added as there is no keywords list after the abstract:

: A keywords list has now been added after the abstract. The keywords included are:

“Keywords: online learning, synchronous learning, asynchronous learning, conventional online course, multi-group analysis”

Copy editing is recommended:

: Thank you for your comment. The manuscript has been thoroughly copyedited to enhance clarity and language accuracy and to ensure full compliance with the PLOS ONE formatting and style requirements.

4. In the Methodology section, within the Participants subsection, lines 226-7, the author stated that the whole number of respondents is 795, showing “313 students reported experience with Syn, 316 with Asyn, and 166 with 227 COC”. Aren’t there cases where some respondents experienced more than one mode? Is it really they experienced ONLY one mode of learning? The whole world experienced the three modes, didn’t they?:

: Thank you for your comment. We addressed this problem.

5. In line 279, there is a typing error:

: Thank you for catching this typo. In the revised manuscript, we fixed this typo in line 279(differences) and 281 (followed). In addition, we thoroughly read the manuscript to find other typos.

6. Titles of table captions are not to end with a full stop which is usually used to end a full sentence.

: Thank you for your comment. All table captions have been reviewed and revised to ensure that they do not end with a full stop, in accordance with PLOS ONE style guidelines.

7. It is highly suggested to have a section titled ’Limitations and suggestions for further work’ before the Conclusion section.

: We thank the reviewer for this helpful suggestion. In the revised manuscript, we have added a new dedicated section titled “Limitations and Suggestions for Further Work” immediately before the Conclusions section to explicitly discuss the study’s limitations and directions for future research.

8. The conclusion section is weak in the sense that it does nor truly conclude the present study. The author is invited to re-work on this bringing the main findings of this study.

: We appreciate this important comment. The conclusion section has been substantially rewritten. The revised conclusion has been substantially expanded to synthesize the main empirical findings, explicitly reflect modality-specific differences identified through latent mean analysis and multi-group SEM, and articulate clearer theoretical and practical implications based on the extended ISSM framework.

9. There is no reason why Figure 2 presented before figure 1 at the end of the paper.

: Thank you for noting this. The figures have been reordered so that Figure 1 is presented before Figure 2, and all figure citations have been checked for consistency.

Reviewer #2:

1. Introduction

Since COVID-19 is no longer a pressing issue in online education, the introduction should focus on the broader development and sustainability of online learning rather than positioning the pandemic as the central rationale.

: We appreciate the reviewer’s insightful comment and agree that online learning should be framed within a broader, long-term perspective. In the revision, following your suggestion, we re-organized the introduction section to more clearly emphasize the need for continued research to support the advancement and sustainability of online learning.

2. Literature Review

Some of the cited references are relatively outdated (e.g., Skylar, 2009; Sun & Wu, 2016), and it is recommended that the authors include more recent studies.

The discussion of synchronous, asynchronous, and conventional online courses remains largely descriptive, without synthesizing the literature to highlight a clear research gap.

The review does not explicitly state the research questions or explain how this study addresses the theoretical or methodological limitations identified in previous work.

: Thanks for the constructive and insightful comments regarding the Literature Review section.

First, in response to the concern about outdated references, we have carefully reviewed and updated the literature by incorporating a range of more recent and relevant studies, while retaining earlier foundational works where theoretically appropriate.

Second, we have substantially revised the Literature Review to move beyond a descriptive overview toward a more integrative and analytical synthesis. The revised sections now explicitly compare prior findings across the three modalities, identify areas of inconsistency and limitation in existing studies.

Third, based on this synthesis, we have explicitly articulated the theoretical and methodological gaps addressed by the present study. In particular, we emphasize the limited application of the Information Systems Success Model (ISSM) to comparative analyses across online learning modalities, as well as the scarcity of studies examining learners’ satisfaction and continuance intention using both latent mean analysis and multi-group structural equation modeling. The research questions are now clearly stated and directly linked to these identified gaps at the end of the Introduction.

Overall, we believe these revisions strengthen the coherence of the Literature Review and clarify how the present study advances prior research by providing a theoretically grounded and methodologically rigorous comparison of synchronous, asynchronous, and conventional online courses.

3. Data Analysis

The logical connection between latent mean analysis (LMA) and multi-group SEM remains unclear. Although the analytical sequence is correct, the authors should specify which group served as the reference in LMA and elaborate on how the three analytical steps are theoretically linked.

While the authors report CFI, TLI, and RMSEA values and mention ΔCFI/ΔTLI/ΔRMSEA thresholds, the actual Δ values and references for these cut-offs are not provided. Including a summary table and relevant citations (Cheung & Rensvold, 2002; Chen, 2007) would enhance transparency.

Finally, please clarify whether the reported fit indices meet accepted benchmarks (CFI/TLI > 0.90, RMSEA < 0.08) and consider reporting standard errors or confidence intervals for path coefficients to strengthen interpretability.

: Thanks for these important comments.

First, we have clarified the logical and theoretical linkage between latent mean analysis (LMA) and multi-group structural equation modeling (SEM) in the revised manuscript. Specifically, we now explicitly state that the conventional online course (COC) group served as the reference group in the latent mean analysis. We also elaborate on how LMA was used to identify modality-level differences in perceived constructs, which was subsequently followed by multi-group SEM to examine whether the underlying structural relationships among ISSM variables operated equivalently across modalities.

Second, to enhance analytic transparency, we now report the actual ΔCFI, ΔTLI, and ΔRMSEA values used in the invariance testing procedure and include them in a summary table. In addition, we have added explicit references to established cut-off criteria for model comparison, including Cheung and Rensvold (2002) and Chen (2007).

Third, we have clarified that all reported fit indices meet commonly accepted benchmarks (CFI/TLI > 0.90, RMSEA < 0.08), and this information is now explicitly stated in the revised Results section.

We believe that these revisions substantially strengthen the clarity, rigor, and transparency of the analytical procedures employed in this study.

4. Discussion and Conclusion

The discussion section interprets the findings largely from an empirical perspective but does not fully draw on the Information System Success Model (ISSM) and its extended variables to explain the underlying mechanisms. The authors are encouraged to explicitly link the observed relationships (e.g., stronger PU→CI and IQ→LS paths in the COC mode) to the ISSM framework to enhance theoretical coherence.

Some explanations (e.g., “teaching presence increases satisfaction in the Syn mode” or “COC performs best overall”) extend beyond what is directly supported by the statistical evidence.

Although the conclusion summarizes the main implications, the discussion remains rather general. The authors could make the implications more specific and actionable, for example, by suggesting how improved information integration and system usability could strengthen the PU→CI relationship observed in the COC mode.

: Thank you for this helpful comment. To address this comment, we have tried to expand our discussion in several ways.

The manuscript has been revised as follows:

• Stronger theoretical integration with ISSM

o The Discussion section has been substantially revised to explicitly interpret the findings within the extended Information Systems Success Model (ISSM).

o Key structural relationships (e.g., PU→CI, IQ→LS, PU→LS, IQ→PU) are now directly linked to the ISSM’s sequential mechanism (quality perceptions → perceived usefulness → satisfaction → continuance intention).

o Modality-specific differences are discussed as variations in how this ISSM mechanism operates across Syn, Asyn, and COC environments, enhancing theoretical coherence.

• Alignment of interpretations with statistical evidence

o All explanations were carefully reviewed to ensure they are fully supported by the empirical results.

o Statements implying causal effects or general superiority (e.g., “COC performs best overall” or unconditional claims about teaching presence) were removed or reframed.

o The revised ‘Discussion’ now interprets findings based on latent mean differences and structural path coefficients, supported by existing literature, without overstating effects.

• More concrete and actionable implications

o The Discussion and Conclusion sections now provide specific, practice-oriented implications rather than general recommendations.

o In particular, the strong PU→CI relationship in the COC modality is discussed in terms of improving the clarity and structural organization of instructional information and enhancing system usability.

o Additional actionable suggestions include incorporating targeted synchronous elements into Asyn and COC designs and providing structured instructional design templates to support sustainable online learning.

We believe these revisions strengthen both the theoretical grounding and the practical relevance of the Discussion and Conclusion sections.

---

## [Decision Letter · Decision Letter 1]

18 Feb 2026

Dear Dr. Shin,

Thank you for submitting your manuscript to PLOS ONE. After careful consideration, we feel that it has merit but does not fully meet PLOS ONE’s publication criteria as it currently stands. Therefore, we invite you to submit a revised version of the manuscript that addresses the points raised during the review process.

**ACADEMIC EDITOR:**  Dear Author,

Thank you for your revised file.

Please revise your paper, highlight the changes, and provide a response letter.

Best,

Ali Derakhshan

We look forward to receiving your revised manuscript.

Kind regards,

Ali Derakhshan

Academic Editor

PLOS One

Journal Requirements:

Additional Editor Comments :

Dear Author,

Thank you for your revised file.

Please revise your paper, highlight the changes, and provide a response letter.

Best,

Ali Derakhshan

Reviewers' comments:

Reviewer's Responses to Questions

**Comments to the Author**

Reviewer #2: All comments have been addressed

2. Is the manuscript technically sound, and do the data support the conclusions?

Reviewer #2: Yes

3. Has the statistical analysis been performed appropriately and rigorously?

Reviewer #2: Yes

4. Have the authors made all data underlying the findings in their manuscript fully available?

Reviewer #2: Yes

5. Is the manuscript presented in an intelligible fashion and written in standard English?

Reviewer #2: Yes

Reviewer #2: Overall, the authors have responded carefully to the reviewer comments and made substantive revisions throughout the manuscript. The major concerns have been adequately addressed, and the revised version shows improved theoretical clarity and methodological transparency. The manuscript is suitable for publication after consideration of the two comments.

1. While the research questions are now clearly stated, the authors may consider explicitly mapping each research question to the corresponding analytical results or subsections in the Results or Discussion sections. Making this linkage more explicit would further improve readability and help readers more easily follow how each research question is addressed empirically.

2. The revised Discussion appropriately focuses on the significant structural relationships. However, a brief reflection on non-significant or weaker paths could further strengthen the interpretation by clarifying why certain expected relationships may not have emerged across learning modalities.

**Do you want your identity to be public for this peer review?** For information about this choice, including consent withdrawal, please see our For information about this choice, including consent withdrawal, please see our Privacy Policy .

Reviewer #2: No

---

## [Author Response · Author response to Decision Letter 2]

24 Feb 2026

Response to Reviewer

Reviewer #2:

1. While the research questions are now clearly stated, the authors may consider explicitly mapping each research question to the corresponding analytical results or subsections in the Results or Discussion sections. Making this linkage more explicit would further improve readability and help readers more easily follow how each research question is addressed empirically.

Response : We appreciate this helpful suggestion. To make the analytical structure more explicit, we revised the manuscript in two complementary ways. First, at the beginning of the Results section, we added brief signposting sentences that clearly indicate how each research question was addressed empirically (RQ1 through latent mean analysis and RQ2 through multi-group structural equation modeling). Second, at the beginning of the Discussion section, we clarified that the discussion is explicitly organized around these two research questions, linking RQ1 to the interpretation of latent mean differences and RQ2 to the examination of modality-specific structural relationships. We believe these revisions improve readability and help readers more easily follow how each research question is addressed across sections.

2. The revised Discussion appropriately focuses on the significant structural relationships. However, a brief reflection on non-significant or weaker paths could further strengthen the interpretation.

Response : We appreciate this insightful comment. In response, we added a brief integrative paragraph in the Discussion that reflects on non-significant or relatively weaker structural paths observed across learning modalities. Rather than reiterating the detailed interpretations of individual paths already discussed earlier, this paragraph provides a higher-level, modality-sensitive perspective that situates these weaker or non-significant relationships within broader instructional and contextual characteristics of synchronous, asynchronous, and conventional online courses. This addition is intended to enhance the interpretive completeness of the Discussion while remaining cautious and fully grounded in the reported results.

---

## [Decision Letter · Decision Letter 2]

13 Mar 2026

Exploring synchronous, asynchronous, and conventional online courses in higher education

PONE-D-25-45310R2

Dear Dr. Shin,

We’re pleased to inform you that your manuscript has been judged scientifically suitable for publication and will be formally accepted for publication once it meets all outstanding technical requirements.

Kind regards,

Ali Derakhshan

Academic Editor

PLOS One

Additional Editor Comments (optional):

Dear Author,

Thank you for the revised file.

Best,

Ali Derakhshan

Reviewers' comments:

Reviewer's Responses to Questions

**Comments to the Author**

Reviewer #2: All comments have been addressed

2. Is the manuscript technically sound, and do the data support the conclusions?

Reviewer #2: Yes

3. Has the statistical analysis been performed appropriately and rigorously?

Reviewer #2: Yes

4. Have the authors made all data underlying the findings in their manuscript fully available?

Reviewer #2: Yes

5. Is the manuscript presented in an intelligible fashion and written in standard English?

Reviewer #2: Yes

Reviewer #2: The authors have carefully addressed the comments raised in the previous review round and have revised the manuscript accordingly. The revisions have improved the clarity of the manuscript and adequately resolved the reviewer’s concerns. I have no further comments and recommend the manuscript for publication.

**Do you want your identity to be public for this peer review?** For information about this choice, including consent withdrawal, please see our For information about this choice, including consent withdrawal, please see our Privacy Policy .

Reviewer #2: **Yes:** tian fengtian feng

---

## [Editor Report · Acceptance letter]

PONE-D-25-45310R2

PLOS One

Dear Dr. Shin,

I'm pleased to inform you that your manuscript has been deemed suitable for publication in PLOS One. Congratulations! Your manuscript is now being handed over to our production team.

Kind regards,

on behalf of

Dr. Ali Derakhshan

Academic Editor

PLOS One